

# RETRIEVAL OF TROPOSPHERIC NO₂ COLUMNS OVER BERLIN FROM HIGH-RESOLUTION AIRBORNE OBSERVATIONS WITH THE SPECTROLITE BREADBOARD INSTRUMENT

Tim Vlemmix[1], Xinrui (Jerry) Ge[1,4], Bryan T. G. de Goeij[2], Len F. van der Wal[2], Gerard C. J. Otter[2], Piet Stammes[3], Ping Wang[3], Alexis Merlaud[5], Dirk Schüttemeyer[6], Andreas C. Meier[7], J. Pepijn Veefkind[3,1], and Pieternel F. Levelt[3,1]

[1]Delft University of Technology (TU-Delft), Delft, The Netherlands
[2]TNO, Delft, The Netherlands)
[3]Royal Netherlands Meteorological Institute (KNMI), De Bilt, The Netherlands
[4](currently at) Wageningen University and Research (WUR), Wageningen, The Netherlands
[5]Royal Belgian Institute for Space Aeronomy (BIRA-IASB), Brussels, Belgium
[6]European Space Agency, Noordwijk, The Netherlands
[7]Institute of Environmental Physics, University of Bremen, Germany

*Correspondence to:* Tim Vlemmix (t.vlemmix@tudelft.nl)

**Abstract.** This paper presents the retrieval method that was developed to derive tropospheric NO₂ columns from UV/VIS spectral measurements obtained with the Spectrolite Breadboard Instrument during the AROMAPEX campaign in Berlin (April 2016). A typical DOAS retrieval approach is followed. For the calculation of air mass factors this study specifically focuses on the impact of the surface reflectance, which varies considerably from pixel to pixel over this urban region. Ground-
5 based aerosol optical thickness measurements are used as prior information. It is shown that retrieved surface reflectance shows good agreement with those derived from Landsat 8 measurements performed on the same day. Furthermore we demonstrate that tropospheric NO₂ columns retrieved for pairs of adjacent pixels are self-consistent in the sense that they do not show a substantial systematic dependence on surface reflectance, in contrast to differential slant column densities. Also some cases are identified to illustrate this on a pixel-by-pixel level. An error budget is provided to quantify the impact of various assumptions
on the accuracy of the retrieval of surface reflectance and tropospheric NO₂ columns. Both in the morning and afternoon flight a NO₂ plume is observed stretching out over Berlin from West to East. Peak values between $15 \cdot 10^{15}$ and $20 \cdot 10^{15}$ molec/cm² are detected, whereas – at much lower spatial resolution – OMI detects peak values between $9 \cdot 10^{15}$ (first overpass) and $4 \cdot 10^{15}$ molec/cm² (second overpass).

## 1 Introduction

Nitrogen dioxide (NO₂) is a reactive gas that is widely considered to be one of the main components of air pollution together with ozone, particulate matter and anthropogenically emitted volatile organic compounds (VOC). NO₂ plays a central role in



(urban) atmospheric chemistry and because of its optical properties it is quite easily detectable with passive UV/VIS remote sensing techniques in contrast to some other major air pollutants. Due to its moderate lifetime it is a suitable indicator for anthropogenic pollution sources involving fuel combustion at high temperatures.

In recent years, satellite measurements by instruments such as OMI, GOME-2 and SCIAMACHY have provided a global picture of abundances and trends in tropospheric $NO_2$ (e.g., Richter et al., 2005; Castellanos and Boersma, 2012). Space-borne observations not only allow to monitor the present state of the atmosphere, but when coupled to air quality models they can also be used to derive emission inventories that quantify source regions on the global and national scale (e.g., Beirle et al., 2011; Ding et al., 2017).

Despite the relevance of observations from space, it has proven to be difficult to match the needs of stakeholders with a focus on city scale air quality monitoring and regulation. The two main obstacles in this respect are the limited spatial resolution of present day satellite observations together with the fundamental inability of passive UV/VIS remote sensing (from a single line of sight) to infer information about the vertical distribution of $NO_2$: high tropospheric column amounts do not necessarily indicate high concentrations at the surface and vice-versa, and this complicates interpretation of tropospheric $NO_2$ columns in terms of air quality. Much is to be expected however from coupling of observational and model data, such as is done within the framework of the European CAMS project http://atmosphere.copernicus.eu/about-cams.

In the last decade a number of instruments for airborne remote sensing of tropospheric $NO_2$ columns were developed, amongst others: AMAX-DOAS (Heue et al., 2005; Wang et al., 2005; Dix et al., 2009; Oetjen et al., 2013), AirMAP (Schönhardt et al., 2015), SWING (Merlaud et al., 2012), ANDI (Lawrence et al., 2015), GeoTASO (Nowlan et al., 2016), ACAM (Lamsal et al., 2017). Also the APEX instrument convincingly demonstrated to have sensitivity to $NO_2$ along with many other geophysical parameters (Schaepman et al., 2015). This development has lead to new insights on tropospheric $NO_2$ at unprecedented high horizontal resolution in urban regions across the globe, e.g. Zürich, Switzerland (Popp et al., 2012); Bucharest, Romania (Meier et al., 2016); Antwerp, Brussels and Liège, Belgium (Tack et al., 2017); the Highveld plateau, South Africa (Broccardo et al., 2017); Leicester, Great Britain (Lawrence et al., 2015); Houston, the US (Nowlan et al., 2016).

This paper describes the first deployment of the Spectrolite Breadboard Instrument (SBI) that was developed in 2015-2016 by TNO. Although the instrument was designed for future application in space, the AROMAPEX campaign that was held in Berlin in April 2016 provided a unique opportunity firstly to test the performance of the instrument and algorithms under representative conditions and secondly to compare the retrieval products of four remote sensing instruments onboard two different aircraft.

In most of the papers on airborne observations listed above, it is recognized that the surface reflectance (SR) is an essential parameter in the retrieval of tropospheric $NO_2$, especially under conditions with low AOT. First of all because the SR strongly affects the sensitivity to $NO_2$ in the lowest part of the troposphere (Fig. 1). This is precisely where the $NO_2$ profile peaks in case of an urban region with many sources. Secondly because the SR does not vary as smoothly as other relevant parameters; instead it exhibits a high spatial variability over urban terrain and therefore it strongly depends on the exact pixel shape and position on the ground. As a consequence, one cannot rely on a-priori data and it must be retrieved for every pixel.



Various approaches are followed to estimate the SR and these can roughly be divided into four categories. The first and most obvious is the direct retrieval of SR from measured radiances (e.g., Popp et al., 2012). Quite similar is the procedure to retrieve SR based on intensity measurements combined with a scaling determined over a reference region with well-known surface reflectance. The latter technique is also known as vicarious calibration. In Meier et al. (2016) a reference region is

used with a relatively low, but well-known surface reflectance taken from the ADAM database. Lawrence et al. (2015) uses a slightly different approach, where, after atmospheric corrections, literature values of SR over dark waters and bright white roofs are used to convert measured intensities to SR. The third category of approaches to retrieve SR relies on instruments that are designed to look not only to the nadir, but also to the zenith (e.g., Oetjen et al., 2013). This methodology avoids certain challenges with respect to radiometric calibration. Nowlan et al. (2016) and Lamsal et al. (2017) adopt a fourth strategy, namely

the use of the MODIS BRDF product.

The SBI used in this study has been calibrated radiometrically and therefore this study uses the first approach described above. This is also preferable, because at the high spatial resolution of SBI no useful BRDF data product is available. Interestingly, the single ideal day of the AROMAPEX campaign coincided with an overpass of the Landsat 8 instrument over Berlin. This allowed a detailed comparison of the retrieved SR, as the spatial resolution of both instruments is quite similar.

The remainder of this paper focuses on the quality of the tropospheric $NO_2$ column retrieval. No $NO_2$ products from other airborne or ground-based sensors are used for validation - this will be part of future studies performed in the framework of the AROMAPEX campaign - but instead several indicators of a consistent $NO_2$ retrieval are discussed.

This paper is structured as follows: first the instrument (SBI) and the campaign (AROMAPEX) are described in Sect. 2. The next section describes the algorithm that was developed for the retrieval of SR and tropospheric $NO_2$ columns. Results are

described in Sect. 4. Interpretation of these results is mostly saved for Sect. 5 (Discussion), because many findings are related partially to model assumptions and practical choices that were made while implementing the conceptual algorithm.

## 2 Instrument and Observations

### 2.1 Instrument

The Spectrolite is a compact, UV/VIS imaging spectrometer that has been developed at TNO for future space application.

The design philosophy (relatively light-weight and low-cost modular system for spaceborne remote sensing of the Earth and atmosphere) and instrument specifications are described in detail in de Goeij et al. (2016). Here we will summarize elements that are essential for the present study.

The Spectrolite Breadboard Instrument (SBI) used in the AROMAPEX campaign has a weight of 8kg (which includes the flight box of 4kg) and a volume of $310 \times 420 \times 190 \mathrm{mm}^3$. It has a spectral coverage from 300 to 490nm with a spectral

resolution <0.3nm for all wavelengths above 400nm. The initial wavelength grid (grating function) was determined using several spectral line sources that illuminated a small integrating sphere. The same measurements were used for determining the slit function. Both the wavelength calibration and slit function width were further optimized for each spectrum measured during the campaign by comparison with a solar spectrum, see Sect. 3.2.



During the entire flight the instrument and detector were kept at a temperature of 25°C with a stability of 0.5°C. A glass transmission window was added in order to prevent contamination from outside air.

With the SBI optical lens system used for this campaign – a microscope objective that will not be part of the final design for deployment of Spectrolite from space – an across-track field of view of 8.3° was achieved. At a flight altitude of 3km above the surface, this corresponds to a width of 435m. This was divided in 74 viewing directions, such that each ground pixel has an approximate across track width of 6m. With an integration time of 0.14s, and the airspeed of around $70\,\mathrm{ms}^{-1}$ this corresponds to a Level 0 pixel size in along track direction of approximately 10m.

Processing of raw data to spectral radiances consisted of the following steps: bad pixel removal, electronic offset and dark current removal, followed by flat fielding and radiometric calibration. With respect to the latter, de Goeij et al. (2016) report an uncertainty of 2-5% assuming no degradation of the integrating sphere (these values apply for un-polarized light). The radiometric calibration also resulted in a pixel to wavelength mapping, which was further refined in the DOAS processing (Sect. 3).

With a SNR for the breadboard set-up of approximately 70 per CCD pixel, considerable spatial binning is needed in order to retrieve $NO_2$: 24 pixels were added in along track direction. This resulted in Level-2 ground pixels of $\sim 6 \times 225\mathrm{m}^2$ in the S-N segments of the flight track and $\sim 6 \times 275\mathrm{m}^2$ for the N-S segments of the flight track, where the airspeed was slightly different. For practical reasons only every second across track spectrum was analyzed (see also the discussion in Sect. 5).

## 2.2 AROMAPEX campaign

The AROMAPEX campaign was organized in Berlin for two weeks in April 2016 in the framework of the ESA Earth Observations program for future mission concepts addressing the relevant scales for retrieving different chemical species by means of different instrumentation and aircraft. Already during previous campaigns within this framework - the AROMAT campaigns held in Romania - such as described in Meier et al. (2016), much was learned about airborne observations of tropospheric $NO_2$ and how these compared to ground-based MAX-DOAS and mobile DOAS observations. The main added value of the AROMAPEX campaign lies in the coordinated observation of tropospheric $NO_2$ with four spectral cameras onboard of two different aircraft: the APEX instrument onboard of a Dornier 228 airplane from DLR, and the AirMAP, SWING and SBI onboard of a Cessna 207-T airplane for atmospheric research from the Free University of Berlin. This allows for the first time a detailed comparison of four instruments and their retrieval algorithms for tropospheric $NO_2$ columns. Ground-based MAX-DOAS and sun photometer measurements, as well as airborne sun photometer measurements allow to further analyze the airborne remote sensing observations of $NO_2$. It turned out that there was only one optimal day during the AROMAPEX campaign period: 21 April 2016 was an exceptional day with no clouds and relatively clean atmospheric conditions: Aerosol Optical Thickness (AOT) $< 0.2$, see Fig. 2a.



## 3   Retrieval Method

### 3.1   Retrieval of Surface Reflectance

With a calibrated sensor such as SBI, it is possible to directly link the effective SR – i.e. the SR value that applies specifically to the center point and shape of a ground pixel as well as to the viewing and solar zenith angle at the time of observation
– to the measured radiance. This however requires that the atmosphere is cloud free, the AOT is known and relatively low such that uncertainties in aerosol optical properties such as single scattering albedo and asymmetry parameter are of secondary importance. Fortunately, such atmospheric conditions were present at the time of the morning and afternoon flights over Berlin on 21 April 2016, see also Fig. 2.

Deriving effective SR from the radiance measurements instead of using an external product (such as MODIS or Landsat
SR) has several advantages. Firstly, one avoids complications due to BRDF-effects that need to be taken into account when using SR obtained with a different sensor under different viewing and illumination conditions. A more in depth-discussion on this matter is included in Sect. 5.2. Secondly, when using an external data set, complications may arise related to re-gridding to the exact shape of the ground pixels observed with SBI. At high spatial resolution observations over urban terrain, such an approach would inevitably lower the precision of the final product.

The Doubling Adding KNMI (DAK) radiative transfer model (de Haan et al., 2017; Stammes et al., 1989; Stammes, 2001) was used to make a look-up table containing simulated monochromatic radiances as a function of the parameters listed in Tab. 1. Under the assumptions described above, measured radiances could be linked to effective SR after multi-dimensional linear interpolation of the look-up table. Fig. 3 illustrates the dependence of radiances on surface reflectance and AOT for a nadir looking sensor at 3.1km altitude. It can be seen that for SR$< 0.35$ (such as observed over Berlin), an increase in AOT leads
to an increase of radiance due to scattering in the atmosphere. The opposite is the case for SR$> 0.35$. This can be understood as follows: for low SR, increasing AOT leads to a higher radiance because scattering in the atmosphere dominates the signal (scattering increases due to the presence of aerosols). For higher SR, scattering at the surface becomes dominant compared to atmospheric scattering. In this regime aerosols have a different net effect because extinction through absorption by aerosols reduces the surface contribution to the radiance.

In order to assess the impact of uncertainties in the AOT on the accuracy of the effective SR and the retrieved tropospheric $NO_2$ columns, the retrieval was carried out for three AOT values: 0.05, 0.10, 0.15. The AERONET sensor stationed at the Free University of Berlin (52.458°N, 13.311°E) shows that from this set of three values 0.10 was most suitable for the morning flight and 0.05 for the afternoon flight. Additionally, retrievals were performed for two different aerosol profiles, both with constant extinction from the surface up to 0.4 and 0.8km respectively. Although this parameter is not crucial for the retrieval
of SR under conditions with low AOT, it is important in relation to the air mass factors derived for the conversion from slant to vertical $NO_2$ columns.

As will be shown in Sect. 4.1 and discussed in Sect. 5.2, various indications were found that scaling of radiances further improves the results. Therefore the algorithm was run both for scaled radiance measurements (+6%) and for non-scaled radiances.





**Table 1.** Overview of relevant parameters characterizing look-up table of radiances and altitude-resolved air mass factors generated with DAK.

| Parameter | Unit | Value(s) |
|---|---|---|
| Wavelength | nm | 440 |
| Sensor altitude | km | 3.1 |
| Surface temperature | K | 291, 300 |
| Surface pressure | hPa | 1015 |
| Surface reflectance | – | 0.01, 0.02, 0.04, 0.08, 0.12, 0.16, 0.20, 0.24, 0.30, 0.40, 0.60, 0.80 |
| Aerosol optical thickness | – | 0.05, 0.10, 0.15, 0.20, 0.25, 0.30 |
| Single scattering albedo | – | 0.98 |
| Aerosol layer (top) height | km | 0.4, 0.8 |
| Zenith angles (solar, viewing) | ° | 0, 4, 5, 8. 10, 12, 15, 20, 25, 30, 35, 40, 45, 50, 55, 60, 65, 70 |
| Relative azimuth angle | ° | 0 |

For low SR (e.g. $< 0.05$) an increase of radiance with 6% corresponds to an uncertainty in the AOT$< 0.05$ or an uncertainty in the SR of about 0.005 (see Fig. 3 and Fig. 15 to be discussed below).

## 3.2 Retrieval of tropospheric $NO_2$ columns

A DOAS analysis (Platt and Stutz, 2008) was applied to the spectra measured with SBI for the retrieval of differential slant $NO_2$
column densities (shortened as DSCD in the text and denoted as $\Delta N^S$ in equations). The fit window spans from 425 to 455nm. First the initial – already highly accurate – wavelength calibration of SBI that was performed in the lab, was further refined for this spectral window (typical shift <0.1nm). A second order polynomial was used to describe the pixel to wavelength map and its parameters were optimized simultaneously with a slit function (modeled as a Gaussian line shape) by comparison with a solar spectrum (Kurucz et al., 1984). Then a DOAS fit was made using differential cross section spectra for $NO_2$ measured at
298K (Vandaele et al., 1998) and Ozone 225K (Bass and Paur, 1985), as well as a Ring spectrum (based on the solar spectrum from Kurucz et al. (1984)). These three spectra were convoluted with the Gaussian line shape that was derived in the previous processing step. Also a third order polynomial was fitted to account for broad band spectral effects such as Rayleigh scattering, scattering and absorption by aerosols and the surface and the broad band impact of absorption by trace gases.

In order to provide a reference spectrum for the DOAS analysis, observations taken over a relatively clean region West
(upwind) of the city center were used. For every across track viewing direction a reference spectrum was obtained over that region for that viewing direction. This approach reduces as much as possible the possibility of systematic effects due to spectral shifts between spectra obtained for different viewing directions. Some practical disadvantages of this procedure are discussed in Sect. 5.





Two corrections are needed before DSCDs that are obtained from the DOAS fit can be converted to tropospheric vertical columns (TVCD). The first accounts for the fact that initially differences in $NO_2$ slant column are obtained with respect to a measurement taken over a reference region. The absolute TVCD over this reference region can only be obtained from external data and for this we use the OMI tropospheric $NO_2$ product (DOMINO). This quantity is combined with a tropospheric air

mass factor in order to obtain an estimate of the slant column over the reference region for the aircraft at 3.1km. A second correction takes into account the fact that the stratospheric $NO_2$ slant column changes between the time when the reference spectrum was recorded and the times of the other spectra in the data set. Also for this estimate, we use the stratospheric $NO_2$ product from OMI.

Tropospheric vertical $NO_2$ columns ($N_{trop}^V$) are derived for every ground pixel observed with SBI through application of the

following formula:

$$N_{trop}^V = \frac{\Delta N^S + M_{trop}^{ref} \cdot N_{trop}^{V,OMI,ref} - \Delta M_{strat} \cdot N_{strat}^{V,OMI,ref}}{M_{trop}}, \tag{1}$$

where $M_{trop}^{ref}$ denotes the tropospheric air mass factor (for airborne observation) over the reference region, $N_{trop}^{V,OMI,ref}$ is the tropospheric $NO_2$ column measured over the reference region by the OMI satellite instrument, $N_{strat}^{V,OMI,ref}$ ($2.3 \cdot 10^{15}$ molec/cm$^2$) is the stratospheric $NO_2$ column over the reference region also derived from OMI observations ($3.1 \cdot 10^{15}$

molec/cm$^2$) and $\Delta M_{strat}$ is the difference in (geometric) stratospheric air mass factor between the time of the reference measurement ($t_1$) and the time when the spectrum-to-be-analyzed was measured ($t_2$). $M^{trop}$ denotes the tropospheric air mass factor (for airborne observation) applicable to $t_2$.

The relative importance of these factors varies over the observed region: over the city center $N_{trop}^V$ is mostly driven by $\Delta N^S$ and $M^{trop}$. Further away from the city, where $\Delta N^S$ values are close to zero, the retrieved $N_{trop}^V$ is mostly determined by

$N_{trop}^{V,OMI,ref}$. The stratospheric correction is of secondary importance, but nevertheless included because both the morning and afternoon flights span a few hours and during this time the stratospheric light path changes.

Air mass factors are calculated for an aircraft at 3.1km altitude using a look-up table of box-air mass factors that was made with DAK and by making assumptions about the vertical distribution of aerosols and $NO_2$. Because accurate profile shape information was not available for both constituents, especially its variability over the city center, air mass factors were

calculated for four combinations of block-shaped aerosol and $NO_2$ profiles (see Tab. 2) in order to assess the uncertainty due to these parameters.

Apart from the viewing and solar zenith angles and the aerosol and $NO_2$ profile shapes (discussed above), the two most important input parameters that are needed to extract the correct box-AMF from the look-up table are the AOT and the SR. Three values for the AOT are considered (see previous section) and three SRs are derived for each ground pixel using the SBI

radiance measurements (Sect. 3.1).

In total 24 retrievals were performed for every pixel (4 profile shape combinations times 3 AOTs times 2 scaling options). The likeliness of each scenario varies, and is discussed in Sect. 5.





**Table 2.** Four combinations of block-shaped $NO_2$ and aerosol profile shapes. Profiles extend from the ground to the height indicated in the table and are set to zero above this height.

| Profile index | NO$_2$ | aerosol |
|---|---|---|
| I | 0.2 km | 0.4 km |
| II | 0.4 km | 0.4 km |
| III | 0.4 km | 0.8 km |
| IV | 0.8 km | 0.8 km |

## 4 Results

### 4.1 Surface reflectance

Figure 4 gives an overview of SR retrieved with SBI, for ground pixels that are observed during both the morning and afternoon flight. Typically, values are retrieved between 0.005 and 0.05, with occasional extremes between 0.1 and 0.2. As can be expected, higher values are observed over urban terrain and lower values over forests and surface waters (see also the figures discussed in Sect. 4.3).

Retrieved SR from SBI is compared with the Landsat 8 SR product (Vermote et al., 2016; Woodcock, 2016) of band 1 (433-453nm), based on measurements that were obtained on the same day. At the time of Landsat overpass (10AM UTC), the SZA was about 42.5°, whereas during the morning and afternoon flight of SBI the SZA varied between $60° - 45°$ and $40° - 60°$ respectively, see Fig. 2a. The viewing zenith angle of Landsat and SBI were similar (close to nadir). Fig. 5 shows that in general the observed along track variability of the SR is quite similar, except for some of the highest peaks. The considerable differences seen for some of the highest peaks in both data sets might be related to the fact that also here exact pixel alignment is crucial and because bright infrastructural elements may have highly non-uniform bidirectional reflectance distribution functions (BRDFs). This will make the comparison more critical to differences in viewing and illumination angle. See also Sect. 5.2 for a more detailed discussion on the SZA dependence of the SR.

Figures 4c and 5c suggest that there is some dependence of the SR on the SZA: larger systematic differences in SR are observed for larger difference in SZA between two times of observation. Based on the morning and afternoon SBI retrievals, this BRDF effect is estimated to be $-0.0003 \pm 0.0001$ per degree difference in SZA (linear fit not shown, but the dependence is consistent with the slope of a line imagined through the median values of the boxplot in Fig. 4). Almost the same value (i.e. within its uncertainty estimate) is found when comparing the SBI and Landsat SR retrievals: $-0.0002 \pm 0.0001$ per degree difference in SZA.

Fig. 5 excludes SR retrievals obtained during a roll movement (turn) of the aircraft. 4.4 these retrievals are not excluded. During these parts of the flight considerable deviation between SBI and Landsat is observed (see also Sect. 4.4). This is understandable not only because in a turn the exact alignment of SBI and Landsat pixels is more prone to inaccuracies and





because of BRDF effects (here related to different viewing angle on the same ground-pixel) but mostly because the look-up table version used for the SR retrieval of SBI only contains radiances for the nadir viewing direction.

Another view on the accuracy of the SBI SR product can be obtained via a statistical approach. Fig. 6 illustrates that retrieved SR for SBI and Landsat have quite similar frequency distributions. The agreement between the two products is best

for AOT= 0.10 in the morning and AOT= 0.05 in the afternoon, which is in line with the AERONET measurement indicating a decrease in AOT during the day (Fig. 2a). Also the figure shows both for the morning and the afternoon that SBI SR is systematically somewhat lower ($\sim$0.005) when no scaling of radiances – as mentioned in Sect. 3.1 – is applied. With a scaling of measured radiances by a factor of 1.06 (+6%) an even better agreement is found with almost no systematic difference, and similar improvement would be found if simulated radiances were lowered by 6%. Possible interpretations of this finding are

discussed in Sect. 5.2.

## 4.2 Differential slant columns

An example of DOAS fit results is shown in Fig. 7. Panel $a$ shows two spectra, one of which is obtained over the reference region. Fig. 7b shows the alignment of differential structures in comparison with the solar spectrum after wavelength calibration and optimization of the width of the modeled slit function. Panels $c,d$ and $f$ show the relative importance of each component

($O_3$, Ring, $NO_2$) to the DOAS fit compared to the residual signal. Clearly ozone is not contributing significantly in this fitting window. The Ring and $NO_2$ spectrum are in this example equally important in their contribution to the total fit (panel $e$). It can be noticed by comparing the magnitude of the residual to the fit result that the effective signal to noise ratio is quite low. As can be expected, the relative magnitude of the noise decreases for longer integration times, see rows 4 and 5 of Fig. 7.

A different way to study the precision of DSCD retrievals, other than by looking at the residuals after the DOAS fit, is by

comparing relative differences (RD) between adjacent pixels (say pixel A and pixel B). In terms of DSCDs, the RD between pixel B and A is defined as:

$$RD_{B,A} = 100 \cdot \frac{\Delta N_B^S - \Delta N_A^S}{\left(\Delta N_B^S + \Delta N_A^S\right)/2}. \tag{2}$$

For low DSCDs, the RD between adjacent pixels are mostly due to measurement noise. For higher DSCDs, the relative differences are increasingly dominated by other parameters that might be different for both pixels, such as differences in SR

between the pixels. Fig. 8 shows the rate at which the absolute value of the relative difference decreases with increasing DSCD. The steepness of this decline provides an indication of the performance of the instrument although it should be emphasized that the steepness is also affected by settings such as exposure time and the number of spectra that are binned in across track direction. The detection threshold can be defined as the $NO_2$ DSCD above which relative differences are $< 50\%$. For the present SBI data set, this is the case for DSCD $> 0.25 \cdot 10^{16}$molec/cm$^2$.





### 4.3 Tropospheric vertical columns

$NO_2$ TVCDs are derived according to the algorithm described in Sect. 3.2 and shown in Fig. 9 and 10. The map of Fig. 9 shows the spatial variability of the tropospheric $NO_2$ columns for the afternoon flight. West of the city region $NO_2$ column densities are very low (both in the morning and in the afternoon) and this can be understood because of the rural character of the Western part of the greater Berlin region and the fact that the wind is coming from this direction on this particular day. A first major $NO_2$ source is the Reuter West power station (600 MW) in the Western part of Berlin and locally the TVCD increases from background levels $< 2 \cdot 10^{15}$ molec/cm$^2$ to a peak vale of $\sim 17 \cdot 10^{15}$ molec/cm$^2$ as observed during the morning flight. A similar range is observed in the afternoon (see Sect. 5.3 for a discussion about the negative values observed in the first part of this flight). Many other $NO_x$ sources are scattered over the city and this leads to a broadening of the plume towards the East as can be seen for each consecutive North-South (N–S) or South-North (S–N) crossing by the plane. Not only a broadening of the plume is observed, also background levels and peak values increase steadily towards the Eastern part of the observed region (Fig. 10). This is in line with OMI observations (Fig. 2d,e) that show considerably higher values to the East of the city region (about $9 \cdot 10^{15}$ molec/cm$^2$ in the morning) compared to the West (about $5 \cdot 10^{15}$ molec/cm$^2$ in the morning). The ratio of 2–3 between peak values of SBI and OMI TVCD retrievals over the same region are quite in line with ratios reported in other studies (e.g. Nowlan et al., 2016; Lamsal et al., 2017; Broccardo et al., 2017) and it demonstrates for urban regions that the spatial scale at which the most pronounced gradients in $NO_2$ TVCDs can be observed is smaller than the size OMI pixels.

Fig. 10 shows that noise is quite dominant for individual viewing directions (the spread is indicated in light grey), especially outside of the plume. However, when looking at across-track median TVCD values (blue line) a quite consistent overall pattern emerges. Worth mentioning are the similarities in the spatial patterns of adjacent N–S and S–N segments. Note that in Fig. 10 for every consecutive crossing of the plume, the spatial pattern appears to be flipped (left to right) with respect to the previous one, due to the change in flight direction. This can be seen more clearly in Fig. 12 (discussed below).

Without additional observations for comparison – this will be part of future studies – it is a challenge to provide evidence for the consistency of the retrieved vertical columns hence for the effectiveness of the retrieval algorithm. It is relevant to investigate the effectiveness of the correction for variabilities in SR on the final $NO_2$ TVCD product, i.e. to demonstrate that the final product does not show a dependence on the surface reflectance. We address this subject below in two different ways: first by looking in detail at a few example cases, then we follow a statistical approach.

An interesting case is provided by a secondary pollution plume that appears to be present somewhat South of the main plume, on the East side of the observed region (see Fig. 9). Fig. 11 and 12 zoom in on a region where a local minimum in tropospheric $NO_2$ column abundances can be observed between the primary and secondary peak. At first it may seem remarkable that this local minimum is not visible in the DSCD product. However, a closer analysis reveals that precisely the absence of this feature in the slant column product and the presence of the feature in the vertical column product together provide a case-based indication of the effectiveness of the retrieval algorithm.

The numbers 1, 2 and 3 in both figures indicate regions where the impact of corrections for surface reflectance on the vertical column densities is substantial. The region indicated with 3 contains a substantial local variation in the surface reflectance: a





relatively bright residential area (3) is preceded by dark vegetation (forest) and followed by a relatively dark lake. It is this fluctuation in the SR, which – through its impact on the air mass factor – reveals a local minimum in the (final) TVCD product that was not visible in the (intermediate) DSCD product. Confidence in the actual existence of this feature is provided by the fact that on the previous flight segment (South-North), a similar local minimum is observed (indicated with the asterisk symbol

in Fig 12c). Here the SR does not vary, thus the minimum is already visible in the DSCDs and propagates to the TVCDs. Also the substantial SR fluctuations indicated by the numbers 1 and 2 – and the resulting corrections that are made through the air mass factor – reduce the considerable along track spatial variability that is observed in the DSCD map. The TVCD map appears to be smoother and therefore more realistic because no smoothing is applied in along track direction (in across track direction median values are used for Fig. 11).

Some more insights into the effectiveness of the algorithm can be obtained by exploiting the relatively high spatial resolution of this data set in a statistical manner. Just like in the previous section, we focus here again on relative differences (RD) in $NO_2$ columns between adjacent pixels (e.g. pixel A and pixel B).

One may expect that the RD in the DSCDs between adjacent pixels are zero on average, and this is confirmed by the histogram shown in Fig. 13. The width of the distribution of relative differences is not only related to the effective signal to

noise ratio of the instrument (and on settings such a exposure time), but also to the variability of the SR because this parameter has significant impact on the DSCDs and may vary substantially from pixel to pixel. We find that RDs in terms of TVCDs are on average smaller (st. dev. 7.4%) than those for DSCDs (st. dev. 10.9%) for pixels with a DSCD $> 20 \cdot 10^{15}$ molec/cm$^2$. This provides a first indication that the correction for SR is effective.

The information contained in the RDs between adjacent pixels can be analyzed even further according to the following

reasoning: there is no relation to be expected between the SR of a ground pixel and the tropospheric $NO_2$ column above it, and consequently TVCD differences between adjacent pixels should not exhibit a systematic dependence on the difference in SR. This reasoning does however not apply to DSCDs, because this quantity is directly affected by the SR. For pairs of adjacent pixels (pixel pairs A and B) where pixels B are brighter than pixels A ($SR_B > SR_A$), one expects for the DSCDs that $\Delta N_B^S > \Delta N_A^S$ (on average) and as a consequence that $RD_{B,A} > 0$. More deviation from 0 is expected for increasing

difference in surface reflectance $\Delta SR_{B,A}$. The positive slope of the blue lines in Fig. 14 shows that this expected behaviour is indeed observed.

A similar analysis applied to TVCDs provides a way to assess the effectiveness of the retrieval algorithm used in this study. In comparison to the systematic impact of SR on DSCDs, we should – for the reason mentioned above – expect almost no systematic dependence on $\Delta SR_{B,A}$ for RDs in terms of TVCDs of adjacent pixels. The red lines in Fig. 14 show that this is

generally the case for the tropospheric $NO_2$ columns that were derived in this study. Furthermore, this figure seems to confirm what was already suggested by Fig. 6, namely that the retrievals based on scaled radiances for an AOT of 0.10 in the morning and 0.05 in the afternoon provide the most consistent picture (see also the discussion on this topic in Sect. 5).



### 4.4 Sensitivities

Fig. 15 shows the impact of critical assumptions (about radiance scaling, AOT and profile shapes) on the retrieved SRs and TVCDs. Not shown is the impact of profile shapes on the SR, because this is negligible for this particular day with very low AOT. The upper panel additionally shows the Landsat SR, as discussed above. It can be seen that the scaling of radiances leads to somewhat higher SR with a median change of about 0.005, with extremes up to 0.01. Assuming higher AOT than 0.05 (default value) leads to differences in surface reflectance of about 0.004 and 0.008 for two and three times higher AOT respectively. Also the impact of roll movements of the aircraft in combination with the (for these particular flight segments wrong) assumption of a nadir viewing zenith angle is visible in Fig. 15 (see the remark on this in Sect. 4.1).

The impact of profile shape assumptions on the AMF is most noticeable over the pollution plume. Here differences up to $1.5 \cdot 10^{15}$ molec/cm$^2$ are seen, which corresponds to around 7–10% uncertainty. Radiance scaling has less impact ($< 1 \cdot 10^{15}$ molec/cm$^2$) for all scenes. The sensitivity to the assumed AOT can be as high as 10%. Note that AOT assumptions are treated consistently: the entire retrieval chain (both SR retrieval and determination of tropospheric AMF) are done for the same assumed AOT.

## 5 Discussion

This section includes a discussion of results presented in Sect. 4 as well as a reflection about choices and assumptions that were made in this study. Although the focus will mostly be on the SR and slant and TVCD retrieval, we begin this section with a discussion of instrumental aspects.

### 5.1 Instrument and data processing

A striking feature of the data set presented above is the narrow swath width (see Fig.9). In this context it is relevant to note that the optical lens system used in this particular experiment was arranged ad-hoc for the AROMAPEX campaign. It will not be included in the future design of the space version of Spectrolite, which will have a field of view of 60° instead of 8.3°. This newer version of the instrument will have a higher throughput as well as a broader swath. Also for this version of SBI a commercial off-the-shelf detector was used with suboptimal pixel sizes. This element will be replaced by a more suitable one in future versions of the instrument.

The data processing chain can also be improved or changed in several ways. For instance: the choice that was made for the large number of across track viewing directions (74) may be defendable – in the sense that the pixel to wavelength mapping and slit function characterization may depend on the viewing direction – but in combination with the optical lens system used for this instrument prototype it leads to narrow, elongated pixels. No thorough testing was done to investigate the impact on the accuracy of the DOAS fit of across track binning, which would lead to more convenient ground pixels with an aspect ratio closer to 1. The number 74 can be reconsidered after data acquisition and is not strictly required by the instrument design.



## 5.2 Surface reflectance retrieval

The algorithm presented in this work quite strongly relies on AOT estimates based on AERONET observations. The importance of this external information is clearly visible in Fig. 6, where for different AOT (e.g. 0.15 instead of 0.05) already considerable differences are found for the retrieved SR. In this case the AERONET station was located somewhat South of the city center, i.e.

South of the pollution plume seen for $NO_2$. Simultaneous ground-based AOT measurements in different parts of the city region (e.g. up-wind, city center, down-wind) would allow a better estimate of the spatial and temporal variability in this important parameter.

A good agreement was found between the SR products of SBI and Landsat. Fig. 6 shows that best agreement was found after scaling of radiances (measured radiance +6%, or similarly, modeled radiance −6%). It should be emphasized however that

this finding may have various possible causes and future experiments are needed to identify the main cause. The most trivial explanation, but not necessarily the most likely one, is that the scaling is needed because of inaccuracies in the radiometric calibration, e.g. because of degradation of the reference used in the lab (as mentioned in de Goeij et al. (2016)) or because of differences between the calibration set-up and the set-up during field deployment. However, so far no such differences were identified.

Polarization effects could also have had an impact on the apparent need for scaling of radiances. These effects may be instrumental (calibration was done for non-polarized light and it is known that the system has a sensitivity to polarization of 20–30%), but also polarization of sunlight scattered in the atmosphere or surface is not included in the radiative transfer simulations. Detailed quantification of instrumental and modeled polarization effects is considered to be beyond the scope of this work, which ultimately focuses on $NO_2$ retrieval.

Another type of explanation for differences between the two SR products could be spatial gradients in the AOT-field that are not taken into account in the retrieval (constant AOT is assumed). This is however quite unlikely: the apparent need to scale measured radiances with a factor $> 1$ implies that the simulated atmosphere is more bright than the measured one for the AOT that is assumed in the retrieval. Not scaling measured radiances would require lower AOT in the radiative transfer simulation in order to match the two (see Fig. 3). This however is quite unrealistic, as the assumed AOT is already as low as 0.05 in the

afternoon.

BRDF effects related to different scene illumination angle (SZA) at the time of Landsat and SBI overpass could also play a role when considering possible explanations for the apparent need for scaling of radiances. Fig. 2 shows that the typical difference in SZA between the times of SBI and Landsat observation was $5° - -10°$. When this is combined with the estimate that was found for the dependence of $\Delta SR$ on $\Delta SZA$ (i.e. $-0.0003 \pm 0.0001$ per degree difference in SZA, see Sect. 4.1),

one may expect a systematic difference between SR from SBI and Landsat of approximately $-0.0015 - -0.0030$. Therefore the BRDF effect may explain 30–60% of the systematic difference in SR (0.0050) that is observed when not scaling radiances, see Fig. (6). Please note that this portion of the systematic differences observed when comparing the SBI and Landsat SR products does not necessarily imply a corresponding systematic bias in the TVCD product: SRs measured at different SZA should not be expected to be the same (because the BRDF effect) and the appropriate SR value should be used for the TVCD retrieval.





This is in principle the one that has been derived from the SBI measurements. It does not have to be equal to the Landsat SR value retrieved for the same ground pixel in case of a substantially different SZA.

Another part of the apparent need for scaling of radiances could be related to one of the input parameters that was specified when running the radiative transfer model: the look-up table was made for an aircraft at 3.1km altitude above sea level,

whereas in practice the aircraft altitude above the city of Berlin varied between 3.05 and 3.1km (A.C. Meier, personal communication) and hence the simulated aircraft altitude was systematically 0.025km too high. Rayleigh scattering by air molecules is the dominant source of scattering affecting the radiance level for solar backscatter measurements in this wavelength range ($\sim$ 440nm) and for conditions with very low AOT (0.05-0.1) and generally low SR (typically 0.03). The impact of 0-0.05km additional air column in the simulations compared to the actual conditions may have an impact of up to 1.5% on the simulated

radiances. Therefore this partially explains the improvement towards the Landsat product of the SRs derived from SBI after scaling of radiances by 6%.

Finally it should be noted that also the Landsat SR product may be biased, for instance because of errors in the atmospheric correction (e.g. for the presence of aerosols).

To summarize: a good agreement was found between SBI and Landsat SR. However, it was noticed that the systematic dif-

ference of 0.005 between the SR products vanishes almost entirely if measured radiances are increased by 6% or, equivalently, modeled radiances are reduced by the same amount. Approximately half of this might be attributed to BRDF effects due to different SZA at the times of measurement. This is a real effect and a bias that should not be corrected for. The remaining part of the bias can have various contributions such as inaccuracies in the radiative transfer modeling (spatial variability of a-priori data such as aircraft altitude and AOT), in the atmospheric correction applied in the Landsat SR retrieval or in the radiometric

calibration of one of the two instruments.

## 5.3 DOAS and tropospheric vertical column retrieval

The DOAS retrieval of DSCDs can be improved in several ways: first of all the signal to noise, and therefore the precision of the DOAS fit, could have been higher for the same computational cost if, instead of processing only every second across track viewing direction, the spectrum of every viewing direction that is not analyzed in the present work-flow (Sect. 3) would

just have been added to the adjacent pixel / spectrum. This would probably not have lead to major implications due to spectral misalignment between the two added spectra, as the corresponding detector pixels are very close. Adding water vapor to the list of absorption cross sections could lead to some changes, although the impact is expected to be small because only substantial differences in water vapor absorption with respect to the reference spectrum will have impact (van Geffen et al., 2015). High spatial gradients in this parameter are not expected over this relatively small region on this clear sky day. Other

improvements could be a more sophisticated parameterization of the effective instrument slit function, such as proposed in Beirle et al. (2017). It should be noted however that despite the room for improvement in the DOAS fitting procedure, we think main conclusions formulated in the next section will hardly be affected. These aspects would be more critical however when trends in tropospheric $NO_2$ column abundances would be investigated over longer time periods or if long-term observations were used to quantify emission sources.





Also it may be possible to optimize further the reference spectrum that is used. Two aspects can be distinguished. The first is that the precision of the $NO_2$ DSCD retrieval will improve if the signal to noise ratio of the reference spectrum is higher. This requires a longer exposure time. Secondly the accuracy of the retrievals is related to the quality of the a-priori information that is available about the tropospheric $NO_2$ column over the region where the reference spectrum was measured.

It can be seen in Fig. 10b, that negative TVCDs (approximately $-1.5 \cdot 10^{15}$ molec/cm$^2$) are retrieved in the first part of the afternoon flight, even after across track averaging in order to suppress the impact of noise. This is an indication of a systematic bias. As can be seen in Eq. 1, low TVCDs are quite likely related to underestimation of the second term in the denominator on the right hand side: $N_{trop}^{V,OMI,ref}$. There are multiple reasons why this estimate based on the OMI measurement could be too low apart from a measurement error induced by noise. One of those is a different timing of the OMI and the SBI measurement:

changes in the $NO_2$ field over the reference region within this period could affect the accuracy of $N_{trop}^{V,OMI,ref}$ in Eq. 1. As can be seen from Fig.2a , this is not the most likely explanation, because the timing of the OMI measurement (2nd overpass at 12:58 UTC) was quite ideal considering the fact that the reference spectra for SBI were obtained early in the flight (e.g. West of the city). A second possibility is a difference in spatial representativeness between the OMI and SBI measurements over the reference region: the SBI reference region represents only a small part of the OMI pixel, see Fig. 2c. However, this

figure shows that about half of the OMI pixel (second overpass of 21 April) overlaps with the city center. This would imply an overestimate rather than an underestimate by OMI with respect to background values in the reference region West of the city. A third possible explanation for an underestimate of $N_{trop}^{V,OMI,ref}$ is the fact that surface albedo used in the OMI retrieval is quite high (0.06) compared to the SRs retrieved by SBI and Landsat (Fig. 5a). An overestimate of the surface albedo corresponds to an underestimate of the $NO_2$ TVCD over that pixel, because of the reciprocal relation between AMF and TVCD. Perhaps this

is the most likely explanation for the relatively low OMI TVCD, together with the impact of noise (a random error).

In order to avoid such uncertainties with respect to the absolute level of the final TVCD product, several aspects can be improved with respect to the present data processing scheme. In the ideal case an airborne reference spectrum is obtained with a high signal to noise, which requires a long exposure time. During this integration period the tropospheric $NO_2$ column below the aircraft should have little spatial variability such that it can be estimated quite well by a MAX-DOAS instrument on the

ground or by an almost simultaneous observation with a high resolution space borne instrument like TROPOMI (Veefkind et al. (2012), launch foreseen in 2017). The advantage of a MAX-DOAS instrument is that the temporal variability of the tropospheric $NO_2$ column that is measured provides an indication of the spatial variability. This improves the estimate of the tropospheric $NO_2$ column in the reference spectrum.

The correction for changes in stratospheric air mass between the time of acquisition of the reference spectrum and any

other spectrum appears to be of minor importance on this day. Nevertheless, this was included in Eq. 1 in order to make sure that the systematic increase of background values observed during the afternoon flight (Fig. 10) was not due to a contribution from the stratosphere. The correction factor for the stratosphere increases with increasing $\Delta SZA$ (compared to reference) from $0°$ to about $15°$ at the end of the afternoon flight. Note that we do not consider here the diurnal cycle of the stratospheric photochemical equilibrium between $NO_2$ and NO, such as in Meier et al. (2016) and Lamsal et al. (2017). Qualitatively this

would lead to a further reduction of TVCDs, which is not substantially different in magnitude from the increase in stratospheric





light path length. Nevertheless it should be emphasized that the increase of background levels observed especially in the afternoon flight is not likely due to a residual signal from the stratosphere, but more likely due to the direction of the wind in combination with the emissions in the Berlin city region (see also Sect. 4.3).

In order to calculate tropospheric AMFs, several assumptions are made. First of all a constant AOT field is assumed because
of lack of accurate information about spatial variability in this parameter at the spatial resolution of SBI. Considering however the fact that a pronounced pollution plume is seen for $NO_2$, the validity of this assumption may be questioned. At the same time, it is reasonable to assume that the spatial gradients in $NO_2$ are substantially higher in an absolute sense compared to those in aerosol optical thickness, given the relatively flat AOT time series often observed by (AERONET) sun photometers: day to day variations are often much higher than variations within a time frame of a few hours, especially on clear sky days
with low AOT.

Another important assumption that was made is the shape of the aerosol and $NO_2$ profile, and with that we refer in this study the top height of a homogeneously mixed layer extending upwards from the surface (simplified profile shape description, see Tab. 2). Also for those layer top heights suitable a-priori information is missing. No spatial (horizontal) gradients in this parameter were assumed, whereas in reality it may be expected that over this urban region the variability of profile shapes is
considerable. This is especially the case on the up-wind side of the city, where few strong sources dominate the profile shape. In that region profile shapes may be found that peak strongly in the first two hundred meters of the atmosphere (e.g. near the Reuter West power station on the down wind side) whereas other profiles in that region represent relatively well-mixed air blown in from the West. Further towards the East (downwind) the sources on the ground will have relatively less impact on the profile shape (more scattered smaller sources, higher background levels). Because of the largely unknown profile shapes, it
was decided to run retrievals for a range of shapes, as discussed before. This gives an indication of the uncertainty due to this parameter and this is quantified a.o. in Fig. 15e.

A final point of discussion is the practical implementation of the algorithm. Although it is quite common to use a look-up table, there are clearly limitations to this approach, especially when the spacing between the nodes increases for practical reasons. This point is illustrated quite clearly in the study by Chimot et al. (2016) on cloud and aerosol retrieval, where
replacement of one (relatively coarse) look-up table with another shows considerably higher accuracy. A follow-up study demonstrates that in terms of data processing speed and accuracy it may be very practical for operational missions to replace a look-up table with a neural network based algorithm (Chimot et al., 2017).

## 6 Summary and Conclusions

The AROMAPEX campaign organized by ESA provided an ideal opportunity to test spectrometers for airborne remote sensing
of tropospheric $NO_2$ that will be deployed in the future amongst others for validation of the sentinel 5 precursor mission. Due to remarkably good weather on 21 April 2016, two flights with excellent test data could be conducted over Berlin. In total 4 spectrometers were used on this day: APEX, AirMap, SWING and the Spectrolite Breadboard Instrument (SBI). A comparison of the retrieval products of these sensors and collocated ground-based observations will be part of a future study by groups



involved in this campaign. The study presented here focuses on the performance of SBI that was developed at TNO (for future application as a space-borne sensor) and the retrieval algorithm that was developed at TU-Delft in collaboration with KNMI. The first results demonstrate that both sensor and algorithm have demonstrated a solid performance. A relatively new element of this study, compared to previous airborne remote sensing of $NO_2$, is the focus on the quality of retrieved SRs and comparison

with an independent data set.

The algorithm to retrieve $NO_2$ tropospheric vertical column densities (TVCDs) relies on the DOAS method to derive $NO_2$ differential slant column densities (DSCDs) and makes use of air mass factors modeled with the radiative transfer model DAK. As an intermediate step, surface reflectances (SRs) are derived from the SBI radiance measurements combined with DAK simulations. These SRs are then used as one of the essential input parameters for the calculation of air mass factors, which

further relies mostly on assumptions about $NO_2$ and aerosol profile shapes. An important element in the retrieval method presented here is the use of external observations of aerosol optical thickness, for which AERONET observations were used from the sensor at the Free University of Berlin.

Retrieved SRs from SBI are compared with Landsat 8 retrievals of the same quantity, based on measurements that were obtained on the same day. Very similar spatial patterns are generally observed. On a more detailed level, we find good statistical

agreement if the a-priori AOT is in line with the AERONET observations (i.e. ~0.10 for the morning flight and ~0.05 for the afternoon flight). It appears however that there is a relatively small shift of (~0.005) in the frequency distributions of the SR from Landsat and SBI. Applying a scaling factor of 1.06 to measured radiances (or the reciprocal of this factor to modeled radiances) further reduces systematic differences observed between Landsat and SBI. This small discrepancy can largely be explained by differences in scene illumination between the time of Landsat and SBI measurements (BRDF effect) and sub-

optimal look-up table settings. More research is needed to fully understand the possible causes of the biases observed (see discussion in Sect. 5).

The DOAS analysis of spectra reveals an $NO_2$ plume stretching out over the Berlin city region from the West to the East in the direction of the wind. In general, the shape of the plume and therefore the location of the pollution can be recognized quite well in the map of $NO_2$ DSCD product. However, the air mass factors that are applied in order to derive $NO_2$ TVCDs show

considerable spatial structure (changes of 30% within a few pixels), driven by variations in SR. A few cases are discussed in detail to demonstrate the impact of this correction on a local scale.

A statistical approach is followed to make an assessment of the consistency of the retrieved TVCDs. We demonstrate that the final $NO_2$ TVCD product shows almost no systematic dependence on differences in SR between adjacent pixels. Also here we find slightly better results after scaling of radiances, but the impact of this scaling is quite small and not driving the main

findings.

Several challenges are identified to determine the $NO_2$ TVCD over the region used to obtain reference spectra needed for the DOAS analysis. For this OMI data is used, but this is not without complications as discussed in Sect. 5. Uncertainties in this parameter may lead to a small offset of the SBI $NO_2$ TVCD product and indeed we see indications in our data (e.g. the negative values in Fig. 10) that for the afternoon flight the $NO_2$ TVCD over the reference region is somewhat underestimated. Follow-

up studies are needed to further quantity this bias and these studies are foreseen within the framework of the AROMAPEX





campaign. It is recommended for future campaigns to select beforehand a region with (a high probability of) background $NO_2$ levels, where airborne reference measurements will be made while a MAX-DOAS instrument is operating from the ground in order to provide an accurate TVCD estimate.

One of the primary future applications of this type of high-resolution TVCD retrievals is identification and quantification

of $NO_2$ sources. In campaigns such as AROMAPEX much can be learned about the quality of the retrieved products, but this is only a first step towards the generation of high-resolution emission maps. The latter requires routine observations and in addition a coupling to a high-resolution transport chemistry model capable of simulating not only surface concentrations but also vertical profiles of tropospheric $NO_2$. In the near future, High Altitude Pseudo Satellites may provide suitable platforms for such routine observations at high spatial resolution.

*Competing interests.* The authors declare that they have no conflict of interest.

*Acknowledgements.* The authors acknowledge ESA and UEFAR for financial support of the AROMAPEX campaign. We thank Thomas Ruhtz and his staff for establishing and maintaining the AERONET site Berlin-FUB from which we use observations in this study. Additionally we would like to express our gratitude to them for kindly providing facilities that were essential for the success of the AROMAPEX campaign. Finally we acknowledge Martijn Schaap (TNO) for taking the primary initiative to involve the Spectolite Breadboard Instrument

in the AROMAPEX campaign.





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





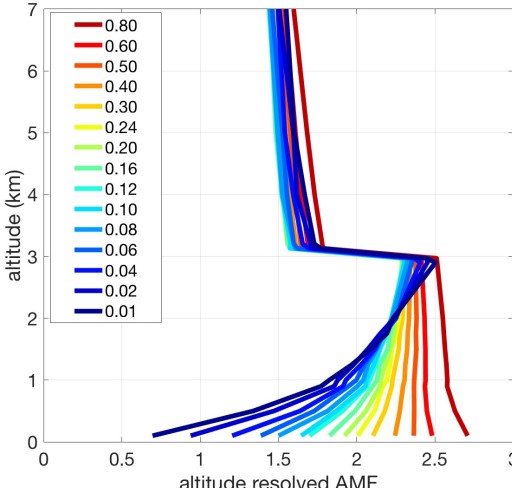

**Figure 1.** Altitude resolved air mass factors simulated for VZA=0° and SZA=50° for an aircraft at 3.1km above the surface. Each line corresponds to a different surface reflectance, see legend.

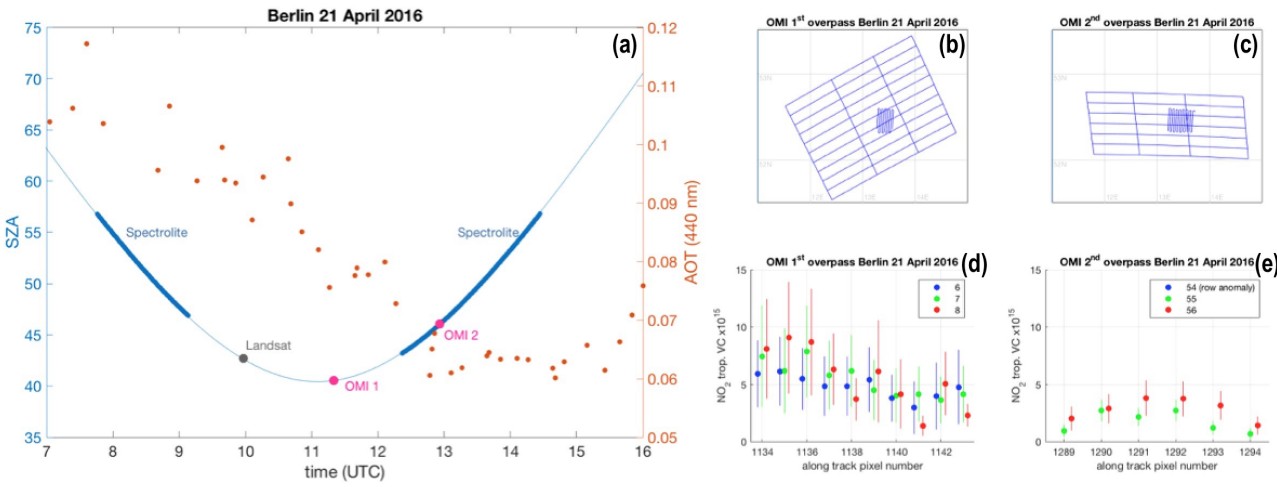

**Figure 2.** (a): SZA variability on 21 April 2016, overpass times of OMI, Landsat and SBI, and aerosol optical thickness measured by AERONET sensor stationed at the Free University of Berlin. (b) and (c): OMI pixel locations and SBI track over the greater Berlin region for the two consecutive OMI overpasses, see (a). (d) and (e): tropospheric $NO_2$ column retrieved for OMI pixels in (b) and (c) as a function of along-track pixel number for three adjacent across track viewing directions (each indicated in red, green and blue). Notice that the lowest along-track pixel number corresponds to the lowest rows in (b) and (c), and the across track pixel numbers increase from the left (blue) to the right (red). In (e), the data from row 54 (the most Western pixels of the second OMI overpass (c)) is not shown because of the OMI row anomaly.





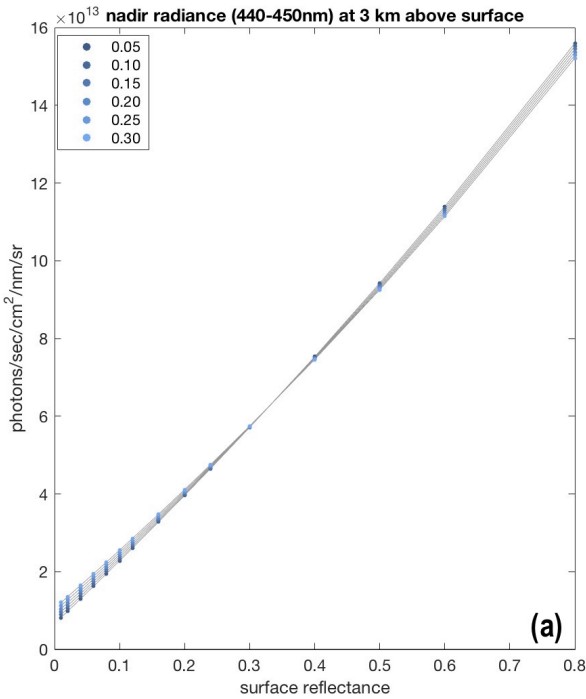

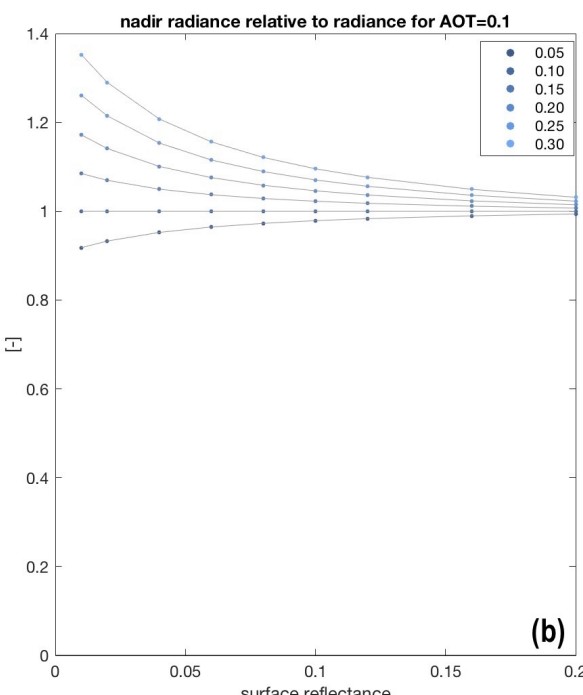

**Figure 3.** (a): nadir radiance (440-445nm) for an aircraft at 3.1 km above the surface as a function of surface reflectance and AOT (see legend). (b): radiance values from (a) relative to radiances for AOT=0.1.





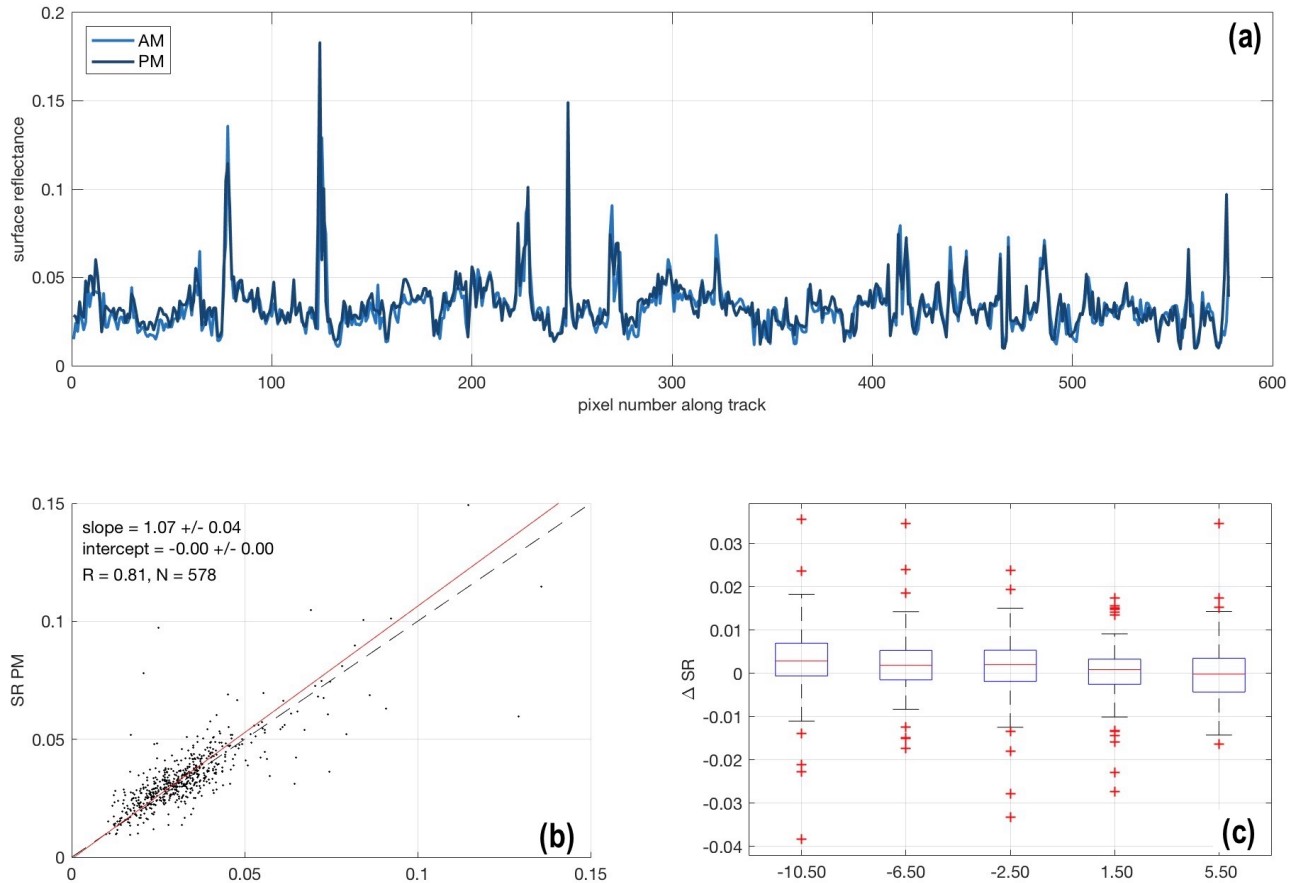

**Figure 4.** (a): Surface reflectances retrieved by SBI in morning and afternoon flight for those areas that are observed during both flights. Across track median values are shown. Flight segments with roll-movements of the aircraft are excluded. (b): scatter plot of data shown in upper panel. (c): difference in SR between morning and afternoon, sorted as a function of the difference in SZA at the times of observation.





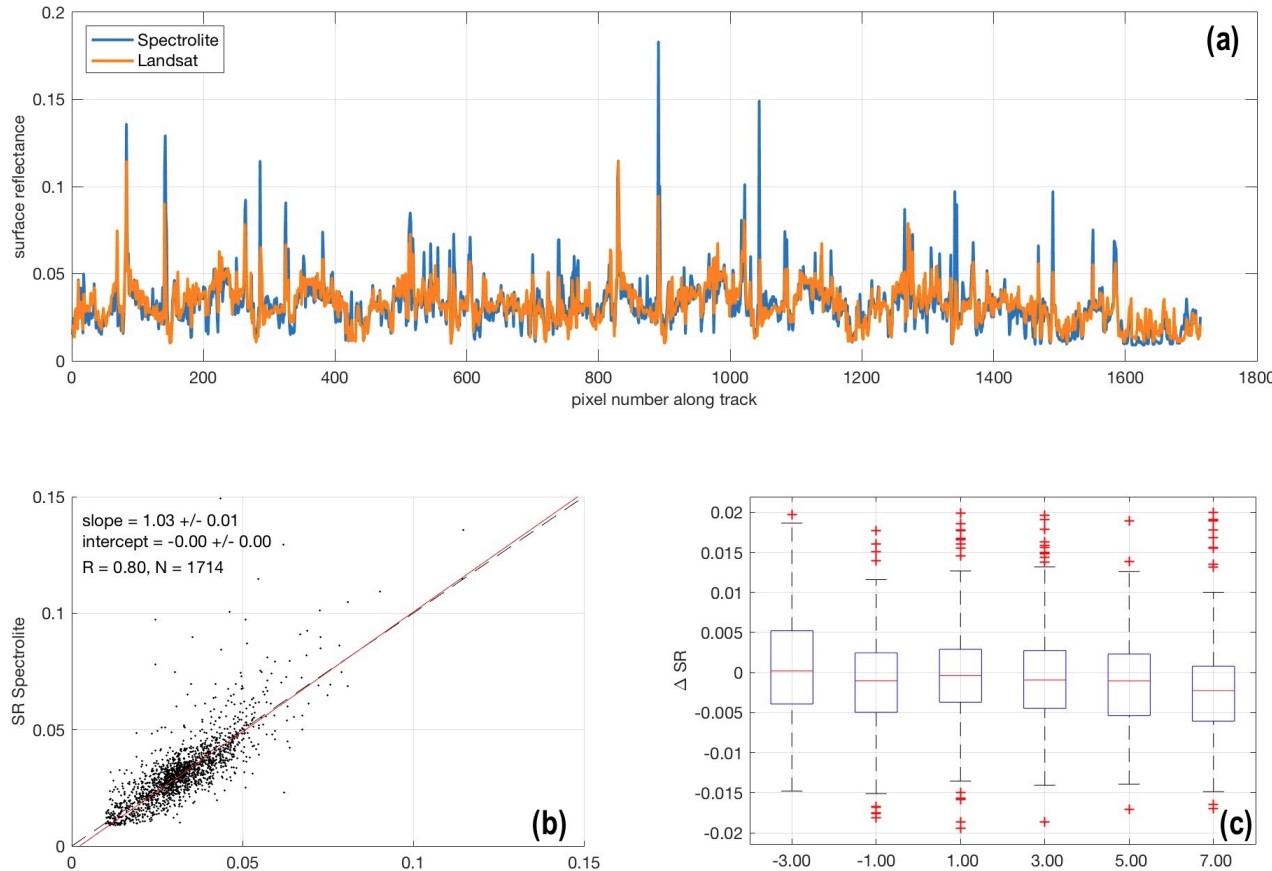

**Figure 5.** (a): Surface reflectances retrieved by SBI and Landsat 8. Retrievals from morning and afternoon flights are combined to one data set. Across track median values are shown. Flight segments with roll-movements of the aircraft are excluded. (b): scatter plot of data shown in upper panel. (c): difference in SR between SBI and Landsat 8, sorted as a function of the difference in SZA at the times of observation.





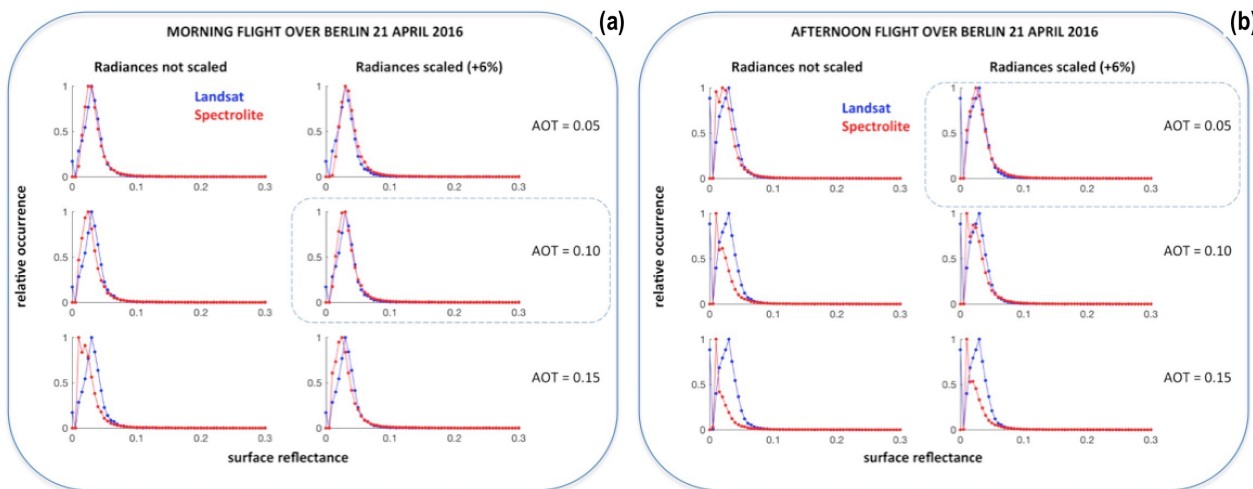

**Figure 6.** Histograms of surface reflectances retrieved with Landsat and SBI for morning (a) and afternoon flights (b). The impact of different a-priori AOT is shown (rows) as well as the impact of scaling of radiances (columns).



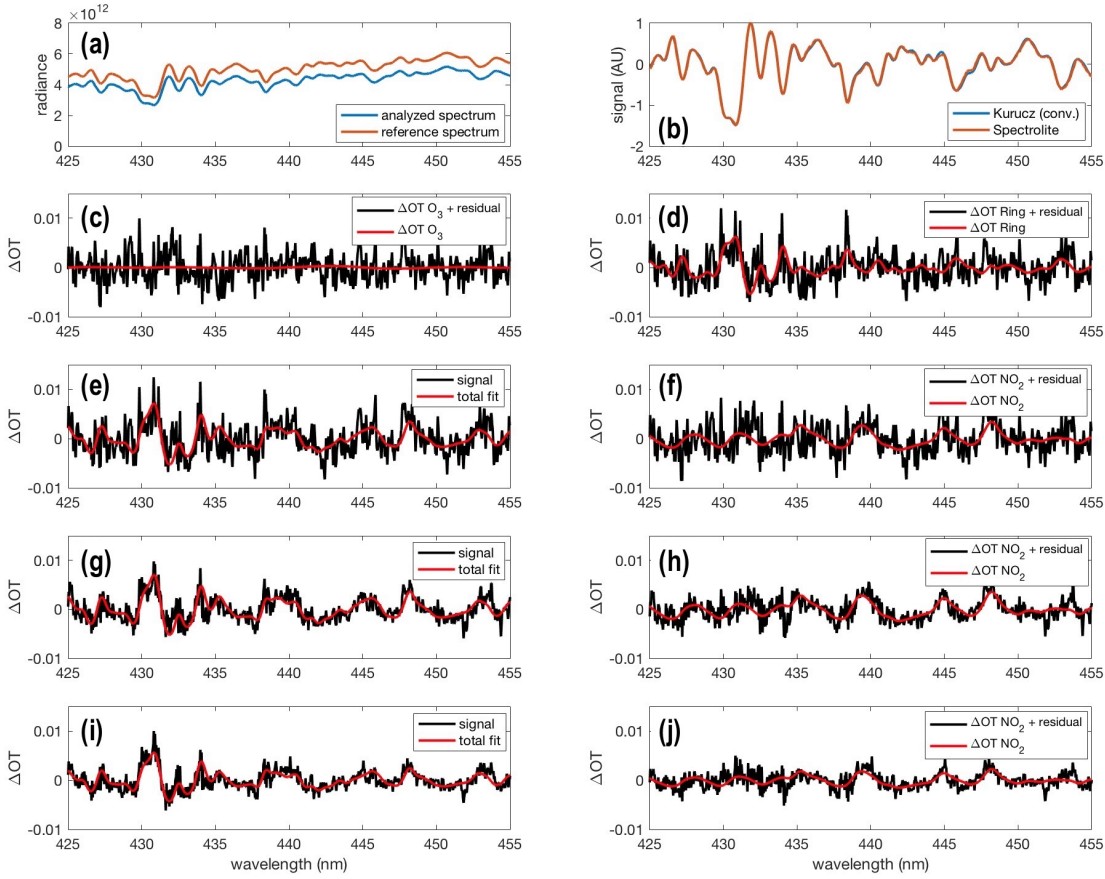

**Figure 7.** Illustration of DOAS analysis. Panels (a-f) are representative for the default settings: spectra of 24 pixels are added in along track direction. An example spectrum as well as the reference spectrum used is shown in panel (a). (b) illustrates the agreement in differential spectral variability after refined estimate of pixel to wavelength map and slit function width. Note that in this panel the blue line refers to the solar spectrum convoluted with a Gaussian slit function. (c) and (d) illustrate respectively the contribution of ozone and the Ring spectrum to the DOAS fit, expressed in units of differential optical thickness ($\Delta OT$). The contribution by $NO_2$ is shown in (f). The total fit ($NO_2+O_3+Ring$) is shown in (e). The fourth and fifth row are similar to the third, but correspond respectively to 5 and 25 times longer integration time.




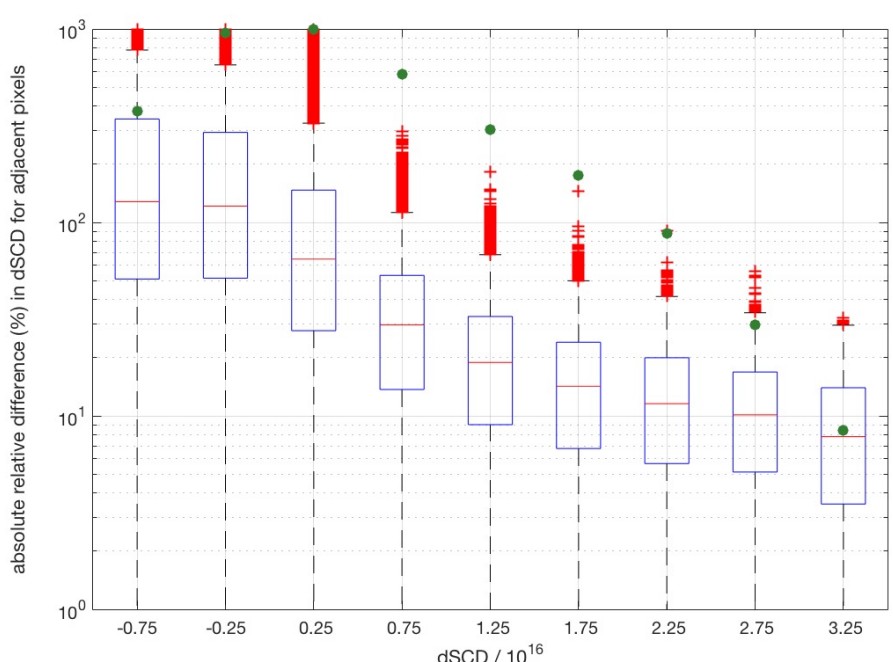

**Figure 8.** Boxplot of the absolute values of the relative differences (%) in $NO_2$ DSCD for adjacent pixels as a function of $NO_2$ DSCD. In green the relative occurrence of each bin is given, normalized to a maximum value of $10^3$.



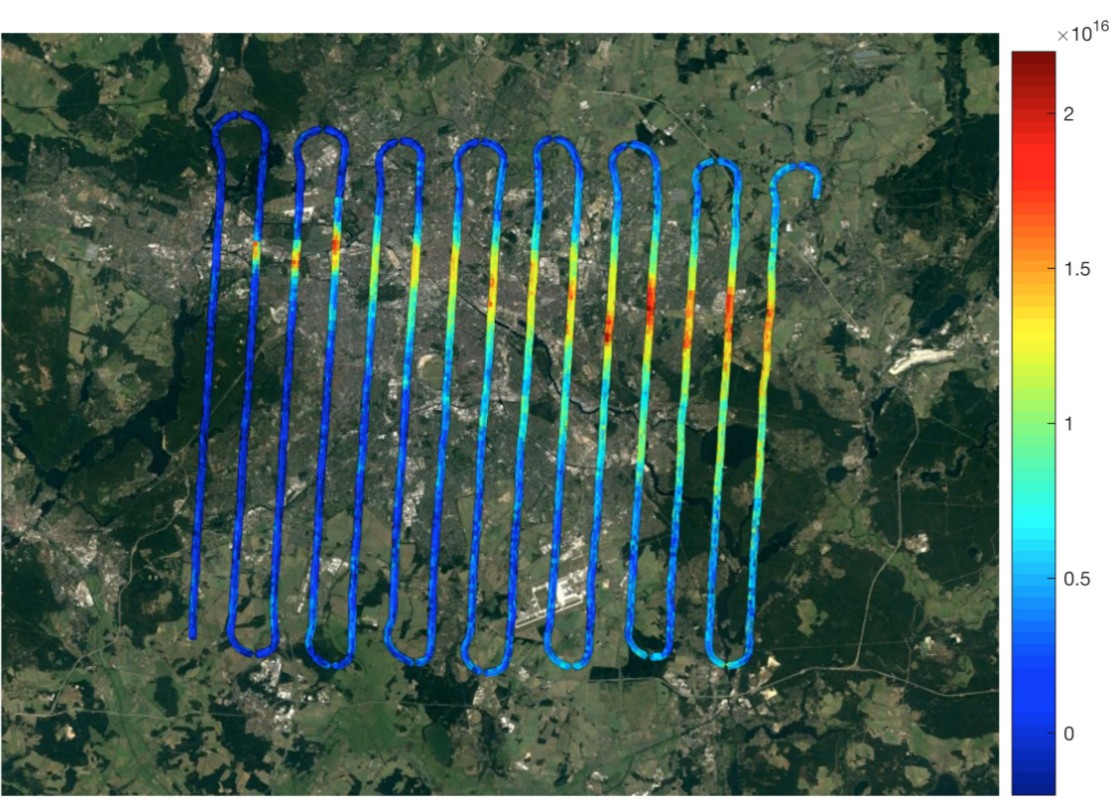

**Figure 9.** Map of tropospheric $NO_2$ columns (in $molec/cm^2$) retrieved for afternoon flight.




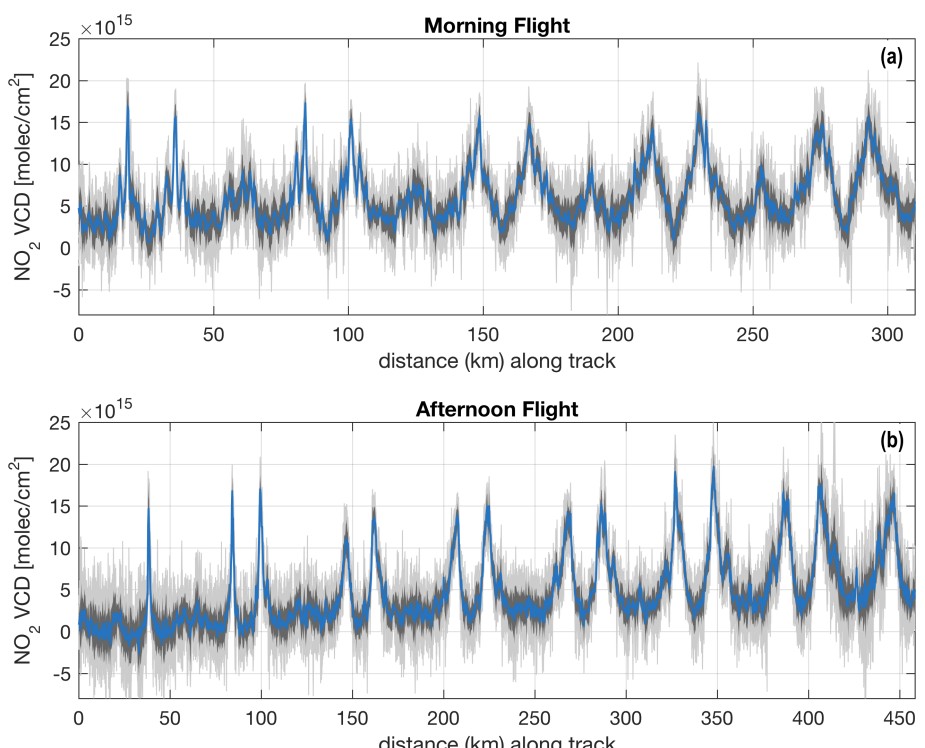

**Figure 10.** In blue: across track median value of tropospheric $NO_2$ columns retrieved in morning (a) and afternoon (b). In light grey: the full range of values retrieved across track. The width of the dark grey region indicates twice the standard deviation of the across track variability.





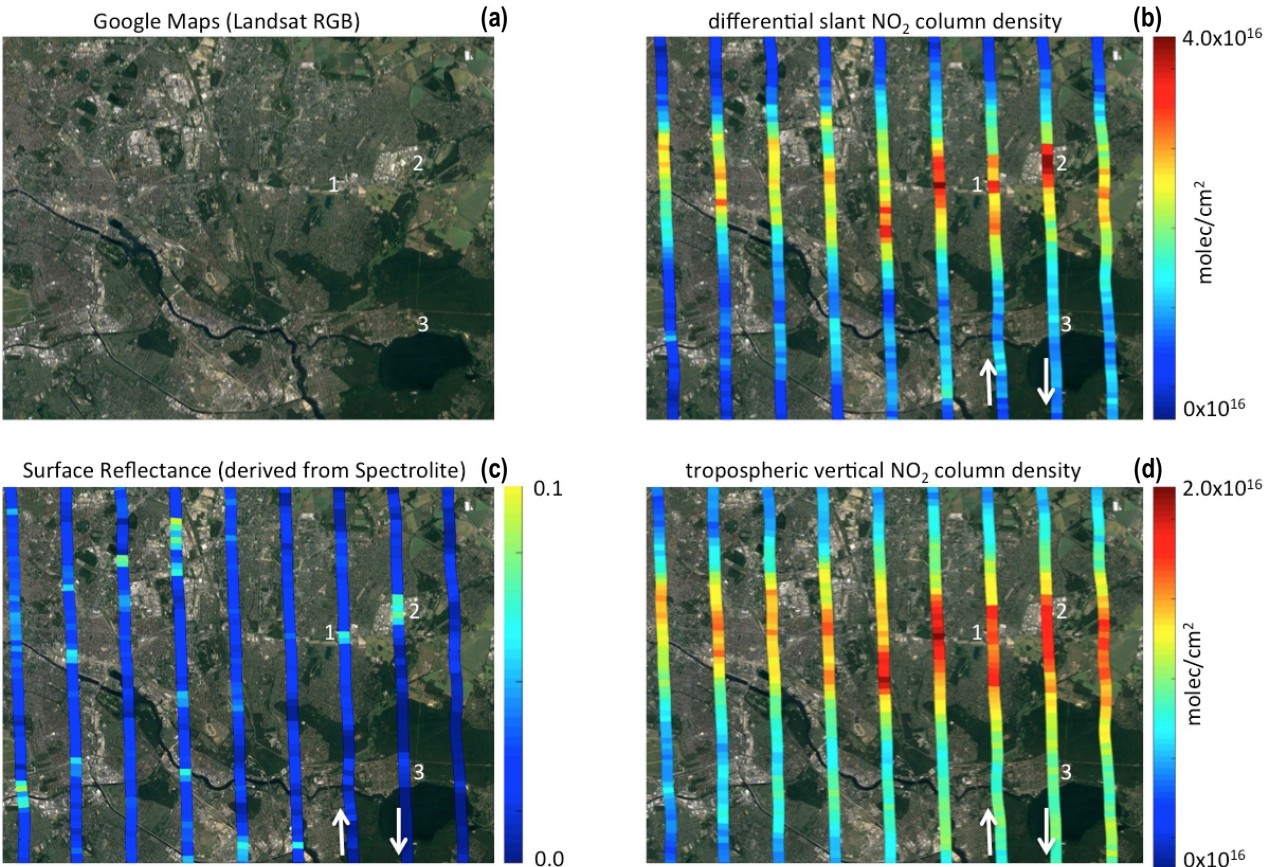

**Figure 11.** Maps of the South-East corner of the Berlin city region. (a): Aerial or satellite imagery of the Berlin region from Google Earth. (b): across track averaged median slant $NO_2$ column retrievals from SBI. (c): across track median surface reflectance. (d): across track median tropospheric vertical $NO_2$ columns. The white arrows indicate the flight direction, which alternates between North-to-South and South-to-North for each consecutive flight segment. The numbers 1-3 indicated in all four panels are mentioned in the text and correspond to the numbers 1-3 in Fig. 12





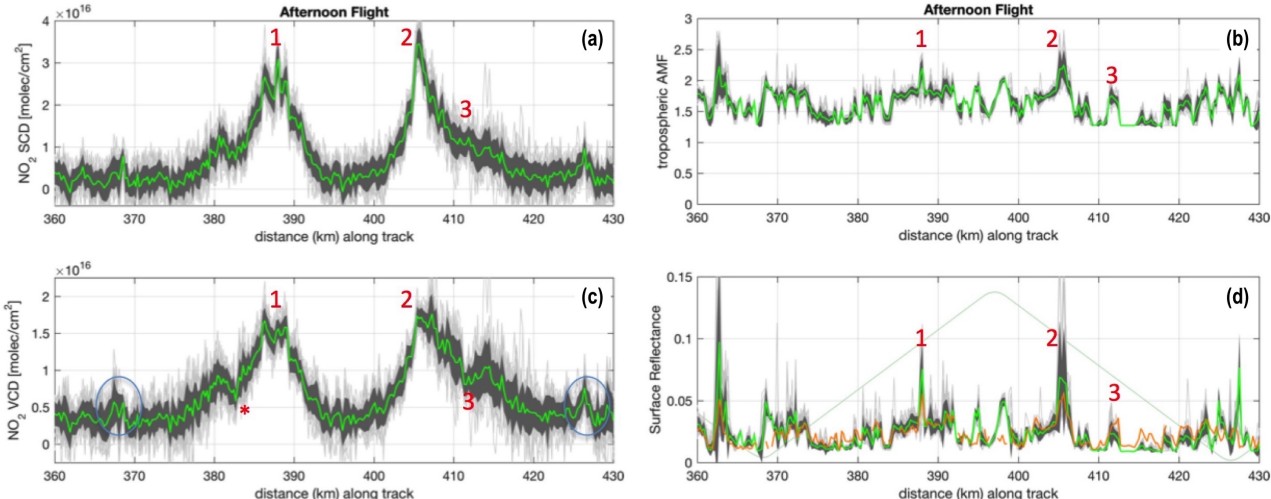

**Figure 12.** (a): NO$_2$ DSCDs for selected section of afternoon flight (see also Fig. 11). (b): corresponding tropospheric air mass factors. (c): NO$_2$ TVCDs. (d): retrieved SR. In this panel the orange line represents the Landsat data. The straight line segments in the background represent the scaled latitudinal position of the aircraft, included in order to demonstrate the impact of roll movements on the SBI SR retrieval (see text). All figures: across track median values are indicated in green, dark grey indicates twice the standard deviation, light grey indicates the total variability.

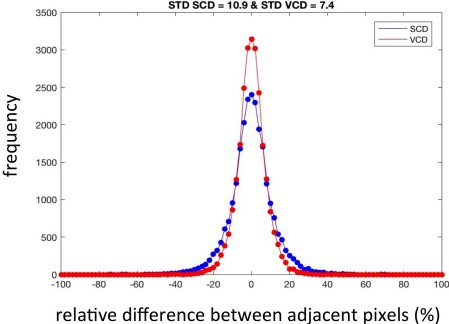

**Figure 13.** Histogram of relative differences (RD) between adjacent pixels for DSCD (blue) and TVCD (red). Only pixel pairs with mean DSCD $> 20 \cdot 10^{15}$ molec/cm$^2$ are considered.




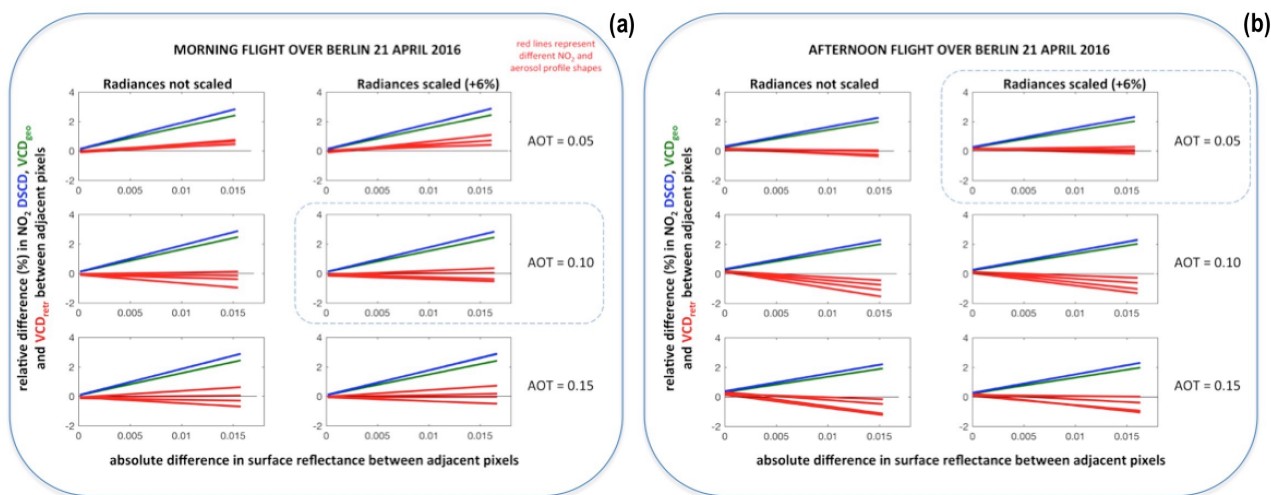

**Figure 14.** Median relative difference in $NO_2$ DSCD and TVCD between adjacent pixels sorted as a function of the absolute difference in surface reflectance of each pixel pair. Panel (a) refers to the morning flight, panel (b) to the afternoon flight. The blue lines represent a linear fit through the median values of $NO_2$ DSCDs. The green lines represent the $NO_2$ vertical columns if a geometric air mass factor is assumed. The red lines refer to retrieved tropospheric $NO_2$ columns after taking into account the impact of surface reflectance. Each red line corresponds to one of the four profile shape combinations (Tab. 2). Similar to Fig. 6, this figure shows the impact of radiance scaling (the two columns of (a) and (b)) and a-priori AOT used (rows).



**Figure 15.** Retrieval results and sensitivities for part of the afternoon flight. (a): Surface reflectance (SR) retrieved by SBI and Landsat. Some turns are indicated with circles, see Sect. 4.1. (b): sensitivity of SR to scaling of radiances (default). (c): sensitivity of SR to assumed AOT. (d): retrieved tropospheric TVCD. (e): sensitivity of TVCD to profile shape assumptions (values in legend refer to layer top height in km, see Tab. 2). (f): sensitivity of TVCD to scaling of radiances. (g): sensitivity of TVCD to AOT assumptions. All panels show across track median values. Grey regions in (a) and (d) show across track variability of the SBI retrieval. The vertical axis of panels (d)-(g) are in units of molec/cm$^2$. Note the changes of the scale on the vertical axis between each panel.