# Peer review of "RETRIEVAL OF TROPOSPHERIC NO2 COLUMNS OVER BERLIN FROM HIGH-RESOLUTION AIRBORNE OBSERVATIONS WITH THE SPECTROLITE BREADBOARD INSTRUMENT"

_Atmospheric Measurement Techniques, 2017_

## Author Comment (AC1) · 25 Sep 2017

**RETRIEVAL OF TROPOSPHERIC** NO2 COLUMNS OVER BERLIN FROM HIGH-RESOLUTION AIRBORNE OBSERVATIONS WITH THE SPECTROLITE BREADBOARD INSTRUMENT**

Tim Vlemmix1, Xinrui (Jerry) Ge1,4, Bryan T. G. de Goeij2, Len F. van der Wal2, Gerard C. J. Otter2, Piet Stammes3, Ping Wang3, Alexis Merlaud5, Dirk Schüttemeyer6, Thomas Ruhtz7, Andreas C. Meier8, J. Pepijn Veefkind3,1, and Pieternel F. Levelt3,1

[revised manuscript text omitted]

---

## Referee Comment (RC1) · Anonymous Referee #1 · 3 Apr 2018

Vlemmix et al. 2018 present a method to derive surface reflectance (SR) and tropospheric NO2 columns from the airborne UV/VIS imaging spectrometer measurements on 21-April-2016 over Berlin, Germany. Calculation of SR is based on the a priori knowledge of the aerosol optical thickness (AOT) and can be applied only when AOT is less then 0.2 at 440 nm. The authors compare the derived SR at 440 nm with Landsat 8 SR (slope = 1.03, R2 = 0.64). Agreement improves when Spectrolite measured radiance is increased by 6%. Differential Optical Absorption Spectroscopy fitting is used to first derive differential slant column densities of NO2 and than using SR, specific profile shapes of aerosols and NO2, and estimation of NO2 column in the reference spectrum to covert to tropospheric columns. Amount in the reference spectrum is es-

timated from OMI NO2 observations. NO2 measurements are shown not to depend on SR. A detailed discussion of uncertainty sources and potential improvements are presented. The paper addresses an important question of estimation of SR by the airborne instrument for NO2 retrieval. The topic is very suited for AMT publication. In my opinion, the scientific community would benefit if the authors divide the current paper in two articles: (1) Feasibility study of surface reflectance measurements from the high resolution airborne UV/VIS imaging spectrometer during AROMAPEX; and (2) Retrieval of tropospheric NO2 columns using various surface reflectance sources during AROMAPEX.

Major comments and recommendations for "Feasibility study of surface reflectance measurements from the high resolution airborne UV/VIS Spectrolite imaging spectrometer during AROMAPEX":

1) The SR retrieval method needs to be described step-by-step, especially the radiance-to-RS linking step. Every assumption should be stated and justification for that assumption should be given. Advantages and disadvantages of this method in comparison with the other radiance based methods should be discussed.

2) More information is needed about the practicality of the surface reflectance method proposed in this paper. The AROMAPEX campaign was 2 weeks in duration but only one day was suitable for surface reflectance measurements (with AOT < 0.2 at 440 nm). Maybe this method could be presented as a "consistency check" method if the required AOT data are available.

3) The authors suggest that the proposed SR method is superior to using Landsat SR data (P5, L9) but than give multiple reasons why their SRs disagree and what it takes to bring them in agreement mainly from the SBI point of view (e.g radiance correction by 6%). Does this mean that the model uncertainties in look-up tables (e.g. different modeled aircraft height vs. actual height; nadir viewing observation geometry for all cases vs. actual, assumption of the AOD spatial homogeneity, etc) make Landsat 8 SR

a better source of SR? Would the radiance scaling (6%) be considered if no Landsat data were available?

4) What errors in tropospheric NO2 columns arise from the difference between SBI and Landsat SRs (this can be a sensitivity study using synthetic data)

5) One of the potential conclusions could be the need of simultaneous airborne aerosol LIDAR measurements to help with SR and NO2 calculations

Major comments for "Retrieval of tropospheric NO2 columns using various surface reflectance sources during AROMAPEX".

1. The biggest disadvantage of the NO2 retrieval is usage of OMI measurements for estimation of NO2 column in the reference spectrum. I am wondering how practical would it be to measure radiances over the same locations with low local emissions at various SZA so modified Langley extrapolation method can be used to derive SCDref.

2. Evaluating effect of SR (SBI derived vs Landsat) on tropospheric NO2

3. Comparison with other independent NO2 tropospheric column measurements is needed to evaluate the effectiveness of SBI retrieval algorithm

Minor comments:

1. Some abbreviations are not spelled out (e.g. DOAS, OMI)

2. P4, L9 Is radiometric calibration performed in flight or do you mean correction?

3. TROPOMI has been launched.

4. Figure 2(a) has no units.

5. Asymmetry parameter is not specified.

---

## Referee Comment (RC2) · Anonymous Referee #2 · 8 Apr 2018

The manuscript by Vlemmix et al. presents observations from an airborne imaging spectrometer to measure NO$_2$ vertical column densities over Berlin during the AROMAPEX campaign. The scientific focus is on studying the effect of surface reflectivity on the conversions of differential slant column densities (DSCD) to tropospheric vertical column densities (TVCD). The manuscript also discusses some more general aspects of the SBI instrument, which was developed in the authors institution. Overall, this is an interesting manuscript that convincingly presents the main point, i.e. the importance of detailed surface reflectivities for NO$_2$ TVCD retrievals. The manuscript clearly fits in the scope of AMT. I found some of the discussion of the instrument, such as Section 5.1, to be somewhat distracting, and I would like to encourage the authors

to focus on their main point when revising the manuscript. I will also point out a few parts of the manuscript that are difficult to read/understand for someone who is not intimately familiar with the concepts used in the UV-vis remote sensing community. I recommend this paper for publication after addressing some minor issues listed in the following:

Page 2 last paragraph: One of the issues I found initially confusing is the definition of surface reflectance, in particular in relation to general terms of reflectivity and bidirectional reflectance distribution function (BRDF). I think it would help the manuscript to define the terms early on, and briefly review the concepts

Page 2, line 31: Figure 1 needs much more explanation. Only absolute specialist will understand this figure. (see also comment on AMF's below)

Page 3, line 10: Define "BRDF"

Page 4, line 14: Why only average along track direction? Would averaging into more square-like pixel not make the interpretation easier?

Line 4, lines 28-31: This is not a sentence. Please reformulate to remove the two colons following each other.

Page 5, Section 3.1: The aerosol extinction profile and its link to AOT is not clear. Is the entire AOT present in the boundary layer, or is some of it above the BL as a background aerosol? How about stratospheric aerosol?

Page 5, lines 32-33: After reading the entire manuscript I understand the motivations for scaling. However, when I first read the manuscript I was thoroughly confused at this point. I think it would help to add a sentence here to explain why it makes sense to scale the radiances. Also, it is not clearly stated which radiances are scaled. I am assuming it is the observed 440nm radiances from the aircraft?

Page 6, lines 9-12: Why were other trace gases that absorb in this wavelength window not included? Most DOAS fits in this window now include $H_2O$, $O_4$, glyoxal, and

sometimes even IO.

Page 6, lines 14-15: How do you know it was a clean region?

Page 7: The concept of the airmass factor needs to be introduced here. AMF's are not universally used in the remote sensing community, and therefore need to be defined. This also applies to height resolved AMF's such as shown in Figure 1. The AMF terms in equation 1 need to be better explained, in particular with respect to the altitudes to which they apply. For example, I am assuming that $M_{trop}^{ref}$ is for downward viewing geometry from the aircraft (once through the troposphere + the lowest 3km from the ground to the AC?). At this point this section can only be understood by an absolute specialist. It would benefit the manuscript greatly to provide a more general explanation. A more general question: When performing the radiative transfer calculation you essentially assume that the surface reflectivity of one pixel applies to the entire atmosphere. However,y our observations are essentially a 3D RT problem. Do you think the reflectance of the surrounding ground, i.e. under the entire slanted light path, has an influence?

Page 7, line 31-32: Can you please explain which of these profiles was ultimately chosen for the TVCD's reported later in the manuscript? How did you select it?

Page 8, Table 2: One would expect that $NO_2$ and aerosol have a very similar box shape, as the profile is determined by vertical mixing, i.e. height of the boundary layer, which influences both equally. Why investigate profiles with different top height?

Page 8 line 22: Something is wrong with this sentence. Please fix.

Page 9, last two paragraphs: Can you provide a more quantitative argument for the approach using adjacent pixel? Is there a statistical basis for defining the detection threshold at DSCD's for which relative differences are smaller than 50

Page 10, line 12: This statement is not supported by Figure 2d,e. Please modify the figure to support your argument (an arrow from east to west, or something similar would

probably be sufficient).

Page 12, Section 5.1: I think this section is not necessary for the manuscript. The fore-optics used for the study has no impact on the main conclusions. The discussion of the choice of binning could be move to Section 2.

Page 13, Section 5.2: It would be interesting to know how well one needs to know AOT and its spatial distribution to improve the observations.

Page 14, line 23-24: I do not remember that processing only every second across track viewing direction was mentioned previously. Either add a section on workflow in section 2, or leave this discussion out.

Figure2: Please increase the fonts on the panels b-e. The gray grid in the panels is not at all readable, even in the PDF version of the manuscript.

Figure 14. I could not find a discussion of $VCD_{geo}$ in the manuscript.